# Chicken Immune Cell Assay to Model Adaptive Immune Responses In Vitro

**DOI:** 10.3390/ani11123600

**Published:** 2021-12-19

**Authors:** Filip Larsberg, Maximilian Sprechert, Deike Hesse, Gudrun A. Brockmann, Susanne Kreuzer-Redmer

**Affiliations:** 1Albrecht Daniel Thaer-Institute, Breeding Biology and Molecular Genetics, Humboldt University of Berlin, Unter den Linden 6, 10099 Berlin, Germany; max.sprechert@web.de (M.S.); deike.hesse-wilting@hu-berlin.de (D.H.); gudrun.brockmann@hu-berlin.de (G.A.B.); 2Institute of Animal Nutrition and Functional Plant Compounds, University of Veterinary Medicine Vienna, Veterinärplatz 1, 1210 Vienna, Austria

**Keywords:** chicken, PBMCs, primary cell culture, immunomodulating compounds

## Abstract

**Simple Summary:**

Knowledge about the modes of action of immunomodulating compounds such as pathogens, drugs, or feed additives, e.g., probiotics, will allow the development of targeted nutrition strategies, prevent infectious diseases and the usage of antimicrobials, and promote the health of animals. To investigate the mechanisms of action of immunomodulating compounds, controlled in vitro systems using freshly isolated immune cells from blood represent a promising alternative to animal experiments. Immune cell isolation from the blood of chickens is a complex and difficult process since the immune cell fractions are significantly contaminated with red blood cells and platelets. To our knowledge, a robust protocol for immune cell isolation from chicken blood and the subsequent cultivation of immune cells is not available. Here, we established a protocol for blood sampling and immune cell isolation and cultivation from chicken blood, which could be applied for the investigation of direct effects of immunomodulating compounds. This protocol, combining different techniques of immune cell isolation, cultivation, and differentiation of distinct immune cell populations, will serve as a potential alternative to animal testing in vivo. By gaining knowledge about the mechanisms of action of immunomodulating compounds, this in vitro model will contribute to promote health and welfare in chicken farming.

**Abstract:**

Knowledge about the modes of action of immunomodulating compounds such as pathogens, drugs, or feed additives, e.g., probiotics, gained through controlled but animal-related in vitro systems using primary cultured peripheral blood mononuclear cells (PBMCs) will allow the development of targeted nutrition strategies. Moreover, it could contribute to the prevention of infectious diseases and the usage of antimicrobials, and further promote the health of the animals. However, to our knowledge, a protocol for the isolation of PBMCs with reduced thrombocyte count from chicken blood and subsequent cell culture over several days to assess the effects of immunomodulating compounds is not available. Therefore, we established an optimized protocol for blood sampling and immune cell isolation, culture, and phenotyping for chicken PBMCs. For blood sampling commercial Na–citrate tubes revealed the highest count of vital cells compared to commercial Li–heparin (*p* < 0.01) and K3EDTA (*p* < 0.05) tubes. Using combined dextran and ficoll density gradient separation, the thrombocyte count was significantly reduced (*p* < 0.01) compared to slow-speed centrifugation with subsequent ficoll. For cell culture, the supplementation of RPMI-1640 medium with 10% chicken serum resulted in the lowest relative cell count of thrombocytes compared to fetal calf serum (FCS) (*p* < 0.05). To validate the ability of the cell culture system to respond to stimuli, concanavalin A (conA) was used as a positive control. The optimized protocol allows the isolation and cultivation of vital PBMCs with reduced thrombocyte count from chicken blood for subsequent investigation of the modes of action of immunomodulating compounds.

## 1. Introduction

Today, there is growing concern about the resistance of pathogenic bacteria against antibiotics, the residual effects of antibiotics in meat products [1], and the public health risk from zoonotic pathogens like *Salmonella* and *Campylobacter*. With the ban of the subtherapeutic usage of antibiotics in farming (Council Regulation EC 70/524/ EEC), immunomodulating alternatives have arisen to improve animal and human health. The immune response to a given stimulus varies between different species [2]. The knowledge of immunomodulatory properties is of high importance, particularly regarding those immunomodulatory compounds with a high potential to improve animal health via improved defence against infection [3]. Therefore, knowledge about the specific modes of action of immunomodulating compounds is needed in order to develop specific diets as alternatives to widely used antimicrobials [4] on farms and to improve the health and welfare of animals, and thereby also humans [5,6,7]. Peripheral blood mononuclear cells (PBMCs) are often used as a cell model to investigate the direct effects of immunomodulating compounds. Methods are well established to isolate PBMCs from the blood of humans [8,9] and animals [5,10,11]. However, in avian species, although it is often performed [5,10,11], it is a complex and difficult process to isolate a population of PBMCs for subsequent cultivation and in vitro assays using common isolation methods, without contamination of nucleated thrombocytes and erythrocytes in the immune cell fractions [11,12,13,14]. Furthermore, erythrocytes, which can be eliminated by red blood cell lysis in mammals, are nucleated in avian species and show a high degree of resistance to common lysis procedures [15]. The majority of erythrocytes are usually eliminated using density gradient centrifugation with ficoll. Thrombocytes can be excluded in further analysis, e.g., by flow cytometry. In contrast to lymphocytes, they express lower levels of the pan-leukocyte marker CD45 [16,17,18]. Moreover, they lack the T-cell marker CD3, the B-cell marker Bu-1a, and the monocyte/macrophage marker Kul-01 [16,19,20], and appear positive for the fibrinogen receptor CD41/CD61 [21] and the CD51/CD61 integrin [17]. Therefore, a dual-labeling approach was established to eliminate thrombocytes [19]. However, as thrombocytes interfere with the quantification of avian leukocytes and can result in shifting percentages of the latter, further markers are needed for the exclusion of thrombocytes [19]. In cell culture, thrombocytes were shown to suffer apoptotic cell death when cultured together with monocytes. The addition of lymphocytes or their soluble factors could delay apoptosis [22]. However, functional ex vivo analyses of, for example, T-cell responses, which are used to determine responses to infections and vaccination in chickens, require an efficient and pure isolation of PBMCs without contaminating thrombocytes [12,20,23]. Furthermore, thrombocytes have been shown to be a major cytokine producer in chickens [24], which would distort the responses of the lymphocytes. The aim of this work was the establishment of a cell culture system with chicken PBMCs to assess the direct effects of potentially immunomodulating compounds on chicken immune cells in vitro, which requires the isolation of a PBMC population with reduced thrombocyte count. Recently, a combined dextran and ficoll density gradient separation was reported to yield large populations of chicken PBMCs without contaminating thrombocytes [25]. Furthermore, a few slow-speed centrifugation approaches have been described previously [12,14,17,26].

However, the isolation of adaptive immune cells from peripheral blood with subsequent primary cell culture over a period of several days for the assessment of the effects of immunomodulating compounds has not yet been described. In this study, different optimization steps were performed to establish an innovative in vitro approach to assess the properties of immunomodulatory compounds. The established in vitro model will help to investigate the modes of action of immunomodulatory compounds such as feed additives and other immune cell stimuli used to improve health in chicken farming and prevent infectious diseases and the usage of antimicrobials.

## 2. Materials and Methods

### 2.1. Animals

Five- to 6-week-old broiler chickens of the commercial layer variety Cobb500 (Cobb Germany Avimex GmbH, Wiedemar, Germany) were used for the establishment of an in vitro cell culture model with chicken PBMCs. The birds were stunned and decapitated. The blood was sampled in tubes containing an anticoagulant. All chickens were fed a starter diet from day 1 to day 14 post hatch, and a grower diet afterwards (H. Wilhelm Schaumann GmbH, Pinneberg, Germany). The ration was fed on an ad libitum basis and water was always available. The light duration was 24 h on days 1 and 2, followed by 16 h/day until sampling. The chickens were kept in groups of approximately 20 chickens in suitable 4 m^2^ pens.

The study was approved by the local State Office for Health and Social Affairs, Landesamt für Gesundheit und Soziales Berlin (LaGeSo, T 0151/19).

### 2.2. Blood Sampling Methods

Different anticoagulants were tested for blood sampling. Commercial 9 mL tripotassium ethylenediaminetetraacetic acid (K3EDTA), lithium heparin (Li–heparin), and sodium citrate (Na–citrate) pre-filled polystyrene tubes (VACUETTE^®^) were used (all from Greiner Bio-One, Kremsmünster, Austria). Additionally, different volumes of 0.5 M EDTA (Carl Roth, Karlsruhe, Germany) in dH_2_O (200 µL, 1 mL, and 8 mL) were tested in 50 mL tubes. Per biological replicate, 30 mL of blood was used for investigation of the best anticoagulant. To test anticoagulants, subsequent cell isolation was performed via combined slow-speed centrifugation and density gradient centrifugation. Isolated PBMCs were resuspended in 5 to 10 mL RPMI-1640 medium (Gibco™, ThermoFisher Scientific, Waltham, MA, USA).

### 2.3. PBMC Isolation Methods

#### 2.3.1. Isolation of PBMCs Using Combined Slow-Speed Centrifugation and Density Gradient Centrifugation

Blood samples were diluted 1:2 with PBS (Gibco™, ThermoFisher Scientific, Waltham, MA, USA) containing 2 mM EDTA (Carl Roth, Karlsruhe, Germany). The samples were mixed and centrifuged for 15 min at 60× *g*. Upon centrifugation, three layers could be observed. The lymphocytes, laid on top of the erythrocyte layer, were swirled up using a dropper and transferred into a new 50 mL tube. The cells were washed once with PBS/EDTA (the tube was filled up with PBS/EDTA to 50 mL), after which the pellet was resuspended in 10 mL PBS/EDTA upon centrifugation, layered 1:2 onto ficoll (Histopaque-1077, Sigma-Aldrich, St. Louis, MO, USA), and centrifuged without a break for 30 min at 400× *g*. The buffer layer was collected at the interface of the plasma and ficoll, transferred to a new 50 mL tube, and washed once by centrifugation at 350× *g* for 10 min. After centrifugation, cells were resuspended in 10 mL RPMI-1640 medium.

#### 2.3.2. Isolation of PBMCs Using Combined Dextran–Ficoll Separation

Blood samples were diluted 1:2 with PBS/EDTA. The diluted blood samples were mixed with 3% dextran in a ratio of 1:0.4 and centrifuged for 20 min at 50× *g*. The upper layer containing the PBMCs was carefully collected and layered onto ficoll in a ratio of 1:2 in a 50 mL tube. After centrifugation without a break for 30 min at 900× *g*, the buffer layer containing the PBMCs was collected, washed twice, and centrifuged for 10 min at 400× *g*. After centrifugation, cells were resuspended in 10 mL RPMI-1640 medium.

### 2.4. PBMC Culture

#### 2.4.1. Cell Counting

The number of isolated vital cells was counted using a Tali^®^ Image-Based Cytometer (Invitrogen™, ThermoFisher Scientific, Waltham, MA, USA). Therefore, 25 µL of cell suspension was mixed with 1 µL of 1 mg/mL propidium iodide (PI) as a viability marker and transferred onto a Tali^®^ Cellular Analysis Slide for measurement.

#### 2.4.2. Cell Seeding

For immune cell culture, cells were seeded in Nunc™ Non-treated T25 EasYFlasks™ (ThermoFisher Scientific, Waltham, MA, USA) at a density of 5 × 10^6^ cells/mL and cultured for several days. For co-culture experiments, cells were seeded in nontreated, flat-bottomed 24-well plates (Eppendorf, Hamburg, Germany) at a density of 1 × 10^6^ cells/mL and cultured from 24 h up to 72 h. All cells were cultured at 41 °C with 5% CO_2_ in RPMI-1640 medium with 2 g/L glucose, 100 U/mL penicillin, 100 µg/mL streptomycin.

#### 2.4.3. Serum Supplementation of Culture Medium

For cultivation of chicken PBMCs over several days, different sera were tested as cell culture supplements. Porcine serum (Sigma-Aldrich, St. Louis, MO, USA), chicken serum (Gibco™, ThermoFisher Scientific, Waltham, MA, USA), or fetal calf serum (FCS, Gibco™, ThermoFisher Scientific, Waltham, MA, USA) was added to the cells cultured in T25 flasks in a concentration of 10% to the RPMI-1640 medium. Immune cells were cultured for 3 days. After 24 h and 72 h, cells were subjected to flow cytometric measurement.

#### 2.4.4. Supplementation of Culture Medium with Additional L-Glutamine

The effect of additional supplementation of 2 mM L-glutamine (Gibco™, ThermoFisher Scientific, Waltham, MA, USA) to the cell culture medium was tested. Therefore, PBMCs with or without additional L-glutamine were cultured in RPMI-1640 medium supplemented with 10% chicken serum for 24 h and subjected to flow cytometry.

#### 2.4.5. Response Capacity of PBMCs towards Immune Cell Stimulants

To validate the cell culture system’s responses to stimuli, concanavalin A (conA, Vector Laboratories, Burlingame, CA, USA) was used as a positive control. Therefore, 5 µg/mL or 10 µg/mL conA was added to the cells cultured in 24-well plates. PBMCs were cultured in RPMI-1640 medium supplemented with 10% chicken serum for 24 h and subjected to flow cytometry.

### 2.5. Immunophenotyping

For immunophenotyping, 1 × 10^6^ cells per antibody staining set were harvested, centrifuged for 10 min at 400× *g*, washed once with cold PBS/EDTA containing 0.05% bovine serum albumin (BSA, Sigma-Aldrich, St. Louis, MO, USA), and stained with different panels of monoclonal antibodies. Therefore, 25 µL of the antibody mix, containing the diluted antibodies in the cold staining buffer PBS/EDTA, was added to the samples. After labeling, the samples were stored on ice in the dark for 30 min. The samples were washed with 600 µL PBS/EDTA/BSA and centrifuged for 10 min at 400× *g*. Subsequently, the supernatant was discarded, the samples were resuspended in 200 µL PBS/EDTA, and analyzed on a flow cytometer. Initially, the most suitable antibody concentration was determined by titration of the respective antibody from 1:25 to 1:400. In this study, immune cells were stained with mouse anti-chicken CD3-Allophycocyanin (APC) (CT-3, SouthernBiotech, Birmingham, AL, USA), CD4-Spectral Red (SPRD) (CT-4, SouthernBiotech, Birmingham, AL, USA), CD28-Phycoerythrin (PE) (AV7, SouthernBiotech, Birmingham, AL, USA), CD8-APC (CT-8, ThermoFisher Scientific, Waltham, MA, USA), CD45-Fluorescein isothiocyanate (FITC) (LT40, ThermoFisher Scientific, Waltham, MA, USA), CD41/CD61-(R)PE (11C3, ThermoFisher Scientific, Waltham, MA, USA), and human anti-chicken CD25-FITC (AbD13504, Bio-Rad Laboratories, Hercules, CA, USA) antibodies. Lymphocytes were gated using forward and sideward scatter by exclusion of debris, erythrocytes, and granulocytes. Dead cells were excluded using 1 µL 4′,6-diamidino-2-phenylindole (DAPI, Sigma-Aldrich, St. Louis, MO, USA) (1 mg/mL). Thereafter, doublets were excluded and at least 20,000 cells in the vital lymphocyte region were acquired on a Canto II (Becton Dickinson (BD), Franklin Lakes, NJ, USA) flow cytometer.

### 2.6. Statistical Analysis

The relative cell count of antibody-positive cells in the flow cytometer was calculated relative to the number of vital lymphocytes. Statistical analysis for blood sampling, PBMC isolation, and PBMC culture was performed using an unpaired Student’s *t*-test. All tests were executed using GraphPad Prism 8.0.2 (GraphPad Software, San Diego, CA, USA). Differences between groups were considered statistically significant at *p* < 0.05.

## 3. Results

### 3.1. Blood Sampling

Blood sampling in commercial Na–citrate tubes revealed the highest count of vital cells (1.11 × 10^8^) compared to commercial K3EDTA (5.37 × 10^7^) (*p* < 0.05) and heparin (2.42 × 10^7^) (*p* < 0.01) tubes (Figure 1). We found no significant difference between blood sampling in commercial Na–citrate tubes and 200 µL 0.5 M EDTA (6.81 × 10^7^).

However, flow cytometric analysis of the relative cell count of CD45-high leukocytes and CD45-low and integrin CD41/CD61+ thrombocytes within the vital lymphocyte population revealed the highest thrombocyte count in blood sampled in commercial Na–citrate tubes after the cell isolation (Appendix A). Since Na–citrate is often used in immunological studies and we also detected the highest total number of live cells after measurement with a Tali image-based cytometer (Figure 1) and BD FACS Canto II (Data not shown), we decided to find a more suitable isolation method to reduce the high thrombocyte count.

### 3.2. PBMC Isolation Method

After blood sampling, the optimal PBMC isolation method was evaluated. In this step, we wanted to decrease the number of thrombocytes and increase the number of leukocytes in blood sampled in commercial Na–citrate tubes. Therefore, the relative cell counts of CD45-high leukocytes and CD45-low and integrin CD41/CD61+ thrombocytes in the vital lymphocyte population were assessed after the isolation of PBMCs using either a combined slow-speed and ficoll, or a combined dextran andficoll separation protocol (Figure 2).

The dextran–ficoll separation did not change the relative cell count of vital leukocytes compared to the slow-speed–ficoll separation method (Figure 2a). Moreover, the mean of the relative cell count of thrombocytes was significantly lower after dextran–ficoll separation (27.23%) compared to the combined slow-speed–ficoll separation (41.19%) (*p* < 0.05) (Figure 2b). The latter was in line with the decrease of cells in the lymphocyte gate (*p* < 0.05, data not shown).

### 3.3. PBMC Culture Conditions

#### 3.3.1. Medium Supplementation with Serum of Different Species

Chicken PBMCs were cultivated in RPMI-1640 medium supplemented with 10% porcine, chicken, or the standard fetal calf serum. The addition of chicken serum was associated with the highest mean of the relative cell count of leukocytes (84.21%) compared to the cultivation with the often-used FCS (54.55%) (*p* < 0.05); interestingly, there was no clear difference to porcine serum (77.04%) (Figure 3a). The thrombocyte count was low in cells cultivated in RPMI-1640 supplemented with chicken serum (13.44%) compared to those supplemented with FCS (43.01%) (*p* < 0.05), but not different to supplementation with porcine serum (21.17%) after 1 day of cultivation (Figure 3b).

After one day of cultivation, the highest number of lymphocytes (Appendix A) and vital cells (Appendix A) was found in PBMCs cultured in RPMI-1640 medium with FCS (60.96%). The relative lymphocyte count was higher compared to culture with porcine (43.16%) (*p* < 0.01) or chicken serum (44.58%) (*p* < 0.05). The live cell count of cells cultivated in FCS (91.16%) was higher compared to those with porcine serum (86.74%) (*p* < 0.05), but did not differ between FCS and chicken serum (89.64%). The relative cell count of leukocytes cultured in RPMI-1640 supplemented with porcine serum (77.04%) was higher compared to cells cultured in medium with FCS (54.55%) (*p* < 0.1). The thrombocyte count was lower in cells cultured in medium with porcine serum (21.17%) compared to FCS (43.01%) (*p* < 0.1). The supplementation of porcine and chicken serum did not differ significantly for leukocytes and thrombocytes.

#### 3.3.2. Medium Supplementation with Additional L-Glutamine

Supplementation with an additional 2 mM L-glutamine to the RPMI-1640 medium with glucose did not affect the viability of cultured PBMCs (Figure 4a). Furthermore, the relative cell count of leukocytes did not change after the addition of L-glutamine, compared to the control without additional L-glutamine (Figure 4b).

#### 3.3.3. ConA as a Positive Control for the Validation of the Response Capacity of the Cell Culture System

To test if the cell culture system was a valid system to examine the direct effects of potentially immunomodulating compounds, the effect of conA, a well-known lymphocyte mitogen which stimulates mainly T-cells, was examined via measurement of CD8+ cytotoxic T-cells (Figure 5a,b) and CD4+ T-helper cells (Figure 5c,d).

As expected, conA treatment increased the relative cell count of CD8+ cytotoxic T-cells (Figure 5a), although the result was not significant. However, at a concentration of 10 µg/mL, conA increased the mean of the relative cell count of CD8+CD25+ activated cytotoxic T-cells from 0.29% to 1.32% (*p* < 0.05) (Figure 5b). For T-helper cells, conA stimulation increased the relative cell count of CD4+ T-helper cells numerically (Figure 5a), but this also did not reach significance. However, looking at CD4+CD25+ activated T-helper cells, conA treatment in a concentration of 10 µg/mL increased the mean relative cell count from 1.40% to 3.54% (*p* < 0.05) (Figure 5d). These effects were visible for two different organs, blood (Figure 5) and spleen (data not shown).

For validation, the effect of 10 µg/mL conA was tested on a higher number of biological replicates (Figure 6). Therefore, the relative cell count of conA-treated PBMCs was assessed by measurement of T-helper cells (Figure 6a), activated T-helper cells (Figure 6b), and all T-cells, except γδ T-cells, via an additional marker, CD28-PE (Figure 6c).

As shown by testing two different conA treatment concentrations on PBMCs (Figure 5), the validation of the effect of 10 µg/mL conA on PBMCs by testing more biological replicates produced similar results (Figure 6 and Appendix A). ConA treatment increased the mean relative cell count of CD4+ T-helper cells significantly from 46.93% to 54.81% (*p* < 0.05) (Figure 6a and Appendix A). Furthermore, conA increased the mean relative cell count of CD4+CD25+ activated T-helper cells from 5.33% to 8.44% (*p* < 0.01) (Figure 6b and Appendix A). Moreover, the mean relative cell count of CD28+ T-cells increased significantly after conA treatment from 76.79% to 88.54% (Figure 6d and Appendix A), indicating T-cell proliferation.

In summary, conA induced T-cell activation and proliferation and can be used as a positive control.

### 3.4. Immune Cell Phenotyping Using Flow Cytometry

To assess the percentage of immune cell subsets, the following gating strategy was used (Figure 7a–d). First, lymphocytes were gated with a SSC/FSC plot (Figure 7a). From the lymphocyte population, only live cells were considered for further analysis (Figure 7b). Doublets were gated out (Figure 7c). Immune cell subsets were analyzed with antibodies labeling leukocytes (CD45-FITC), thrombocytes (CD41/CD61-(R)PE), T-helper cells (CD4-SPRD), cytotoxic T-cells (CD8-APC), T-cells (CD28-PE and CD3-APC), and activated T-cells (CD25-FITC) (example CD45-FITC; Figure 7d and Table 1).

All antibodies used in this study were titrated and evaluated by mean fluorescence intensity (MFI) to obtain optimal concentrations for the staining assays (Table 1 and Appendix A). The following antibody sets were used: CD45-FITC, CD41/CD61-(R)PE, CD3-APC; CD4-SPRD, CD28-PE, CD25-FITC; CD8-APC, CD28-PE, CD25-FITC.

## 4. Discussion

To investigate the mode of action of immunomodulating compounds such as pathogens, drugs, or feed additives, e.g., probiotics, primary cell culture systems are inevitable and represent a good alternative to in vivo models. Our aim was the establishment of an in vitro cell culture model with chicken PBMCs, without contaminating thrombocytes, to assess the properties of immunomodulating compounds, especially feed additives. The study will help to increase knowledge about precise mechanisms of action, which will allow the development of targeted nutrition strategies, prevent infectious diseases and the usage of antimicrobials, and further improve the health in poultry production. Chicken PBMCs are often isolated using common methods such as density gradient centrifugation using ficoll, which result in the isolation of immune cells contaminated by nucleated thrombocytes and erythrocytes [11,12,13,14]. Therefore, the isolation of PBMCs without either nucleated thrombocytes or erythrocytes is complex and difficult in avian species. Concerning this matter, a dextran–ficoll separation method was recently published [25]. However, PBMCs were not cultivated for a longer period after the isolation. Here, we present a robust protocol for longer cultivation of chicken PBMCs, which will enable functional in vitro studies in chicken PBMCs.

### 4.1. Blood Sampling

Our data suggest that, despite a higher relative thrombocyte and a lower relative leukocyte count, the cell number and viability in Na–citrate tubes was highest compared to commercial K3EDTA and Li–heparin tubes, as well as to 200 µL 0.5 M EDTA. In other species, it has been shown that a slightly purer population of PBMCs is obtained by using EDTA as an anticoagulant compared to heparin [27]. Furthermore, for the purification of mononuclear cells from other sources than peripheral blood, heparin was shown to promote clumping and pre-activation of unstimulated control cells [28]. Citrate as an anticoagulant may result in better quality of RNA and DNA compared with other anticoagulants and furthermore produce a higher yield of mononuclear cells, which is consistent with the results of the current study. Li–heparin, which revealed the highest leukocyte and the lowest thrombocyte count in this study, is reported to affect T-cell proliferation and to bind to many proteins. RNA yields from EDTA-treated blood have been shown to be higher compared to heparin-sampled blood [29]. Furthermore, EDTA was shown to affect PBMCs by a progressive and irreversible loss of antigen-specific lymphoprolerative responses when PBMCs were exposed to EDTA for a longer time period [30]. Therefore, ethyleneglycol-bis-(beta-aminoethylether)tetraacetate (EGTA) was suggested [30]. However, in the current study, K3EDTA revealed lower relative lymphocyte and leukocyte counts and a lower count of total isolated vital cells. Taking these findings from the literature and our results into account, we choose Na–citrate as the best choice for anticoagulation.

### 4.2. PBMC Isolation Method

The reduction or elimination of thrombocytes, which are a large part of chicken PBMCs, is of high importance in immunomodulatory studies, since chicken thrombocytes have been shown to play roles in inflammation and antimicrobial defence [24,31,32]. Initially, we used a combined slow-speed centrifugation with subsequent density gradient centrifugation using ficoll, which was modified according to Viertlboeck and Göbel (2007) [17], Lavoie et al. (2005) [14], Sundaresan et al. (2005) [26], and Gogal et al. (1997) [12]. In comparison to those studies, we were not able to reduce the thrombocyte and erythrocyte counts in our samples (Figure 2). Other studies used only density gradient centrifugation to isolate PBMCs from chickens, without considering the thrombocyte count [10,27,33,34]. However, in this study, a method by Jergović et al. (2017) [25] was additionally tested. The method included the use of dextran, a slow-speed centrifugation step and a density gradient centrifugation step afterwards. Dextran, a high-molecular polysaccharide, is often used for the purification of neutrophils [35]. Other studies used 1% methylcellulose instead of dextran [36,37]. Our results showed a significant decrease of the relative cell count of thrombocytes by about 14% when using the dextran–ficoll separation method (Figure 2b). However, compared to the study of Jergović et al. (2017), where the thrombocyte count was about 2.85% compared to 96.9% of PBMCs, we found a thrombocyte count of 27.23%. However, the thrombocyte count was lower compared to the slow-speed centrifugation and ficoll separation method, in which we obtained 41.19% thrombocytes. However, the leukocyte count remained unaffected in our study (Figure 2a). One possible explanation is the dextran used in the different studies, which was provided by different companies. Furthermore, we centrifuged for 20 min after the addition of dextran, because we could not discriminate the phases after 10 min of centrifugation. In the study of Jergović et al. (2017), the centrifugation step was 10 min. All in all, we could reduce the thrombocyte count in our samples and start the cell culture with a low number of thrombocytes when we used the combined separation method based on the addition of dextran, a slow-speed centrifugation step, and a density gradient centrifugation.

### 4.3. PBMC Culture Conditions

We tested sera from different species as cell culture supplementation. Fetal calf serum is very often used even for nonbovine species such as mice, humans, and chickens [38,39,40,41]. Since we detected an increased number of thrombocytes in the cell culture with RPMI-1640 medium supplemented with FCS compared to supplementation with chicken serum, we do not recommend the addition of FCS. Thrombocyte cytokine expression was reported up to 18 h in culture [42], which could interfere with assays that are carried out to investigate immunomodulatory functions. DaMatta et al. [22] and Lam [43] showed that thrombocytes cultured in DMEM with 10% fetal bovine serum displayed cytoplasm and chromatin condensation and were suggested to suffer an apoptotic cell death in culture of about 70% after 24 h and 85% after 48 h of cultivation. Apoptosis of thrombocytes was additionally reported by Kaspers and Kaiser (2014), who showed that cells die by apoptosis after 48–72 h [44]. Despite the fact that thrombocytes are active for a short period, our aim was to reduce the thrombocyte count as much as possible to ensure the least interference with subsequent immunomodulatory assays. However, in our study, thrombocytes were not cultured under agitation to prevent adherence. Interestingly, there seemed to be a difference between the adherence in FCS-supplemented medium to the adherence in porcine- and chicken-serum supplemented media. Whether the serum influences the adhesion of thrombocytes still needs to be elucidated. In addition to the reduced thrombocyte count in cultures supplemented with chicken serum, we found that supplementation with chicken serum resulted in higher counts of vital leukocytes. Therefore, we showed that chicken serum is the most suitable supplement for long-term chicken PBMC cultures with reduced thrombocyte counts.

As L-glutamine is very unstable in medium, the supplementation of additional L-glutamine has been reported in different studies for PBMC culture in RPMI-1640 medium [10,45,46,47] in different concentrations. In our study, we added L-glutamine in a concentration of 2 mM to the cell cultures. We could not detect differences for cell viability or cell counts between media with and without additional supplementation of L-glutamine. Hence, we did not include the supplementation of additional L-glutamine in our protocol.

It is of high importance to validate the response capacity of cultured chicken immune cells to a stimulus. We chose conA, since it is a well-reported T-cell mitogen [12,13,48,49,50]. We observed a clear effect on the proliferation and activation of CD8+ cytotoxic T-cells (Figure 5a,b) and CD4+ T-helper cells (Figure 5c,d) in the presence of conA. Furthermore, we validated the effect on T-cell activation and proliferation (Figure 6a–c and Appendix A). In a study by Alvarez et al. (2020), it was shown that chicken splenocytes have a low viability after conA treatment in a concentration of 1 µg/mL for 3 days [51]. In fact, the cell viability decreased also in our study after treatment with 10 µg/mL conA, but not with 5 µg/mL. However, we aimed initially to validate the responsiveness of the chicken PBMCs in culture as a positive control. Therefore, we were able to validate the presented protocol for a system to culture chicken immune cells.


**Summarized brief protocol:**
Sample blood in Na–citrate VACUETTE^®^ tubesDilute sampled blood 1:2 with PBS containing 2 mM EDTAAdd 3% dextran solution in a ratio of 1:0.4Centrifuge 50× *g* 20 minCollect upper phaseOverlay Histopaque-1077 1:2 with upper phase from step 5Centrifuge 900× *g* 30 minCollect PBMCs at interphase btw. serum and Histopaque-1077 and transfer into a new collection tubeWash twice with PBS/EDTA, centrifuge 400× *g* 10 minResuspend PBMCs in RPMI-1640 supplemented with 10% chicken serum, 100 U/mL penicillin and 100 µg/mL streptomycinCount cells and adjust to 1 × 10^6^/mL up to 5 × 10^6^/mLCulture cells at 41 °C and 5% CO_2_


## 5. Conclusions

After several optimization steps, we established a valid in vitro cell culture system to assess the direct effects of potentially immunomodulating compounds.

Here, we tested and optimized blood sampling, PBMC isolation, PBMC culture, and immune cell phenotyping of chicken PBMCs using monoclonal antibodies. This cell culture system will help to evaluate and understand the underlying mechanisms of the immunomodulatory properties of potentially immunomodulating compounds, e.g., feed additives, which could serve as potential alternatives to antibiotics, and may further serve as an alternative to animal testing in vivo. Besides testing feed additives, immunomodulation by challenges with pathogenic bacteria like *Salmonella* or *Campylobacter*, viruses, or particular drugs could be tested in our established chicken immune cell assay to model adaptive immune responses in vitro.

## Figures and Tables

**Figure 1 animals-11-03600-f001:**
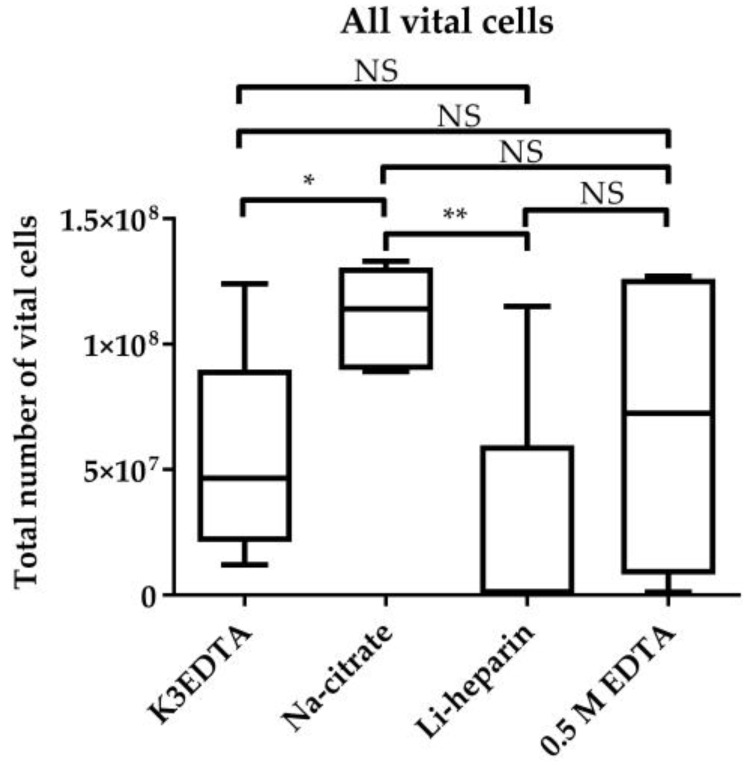
Influence of the anticoagulants K3EDTA, Na–citrate, and Li–heparin as well as 200 µL 0.5 M EDTA on cell survival. After PBMC isolation, chicken immune cells were counted with a Tali image-based cytometer and the viability was assessed using propidium iodide (PI). Five biological replicates are displayed. All bars represent one experiment. A box-and-whisker plot is displayed. Significance is shown as **; *p* < 0.01; *, *p* < 0.05. NS: not significant. Significance was analyzed using an unpaired Student’s *t*-test.

**Figure 2 animals-11-03600-f002:**
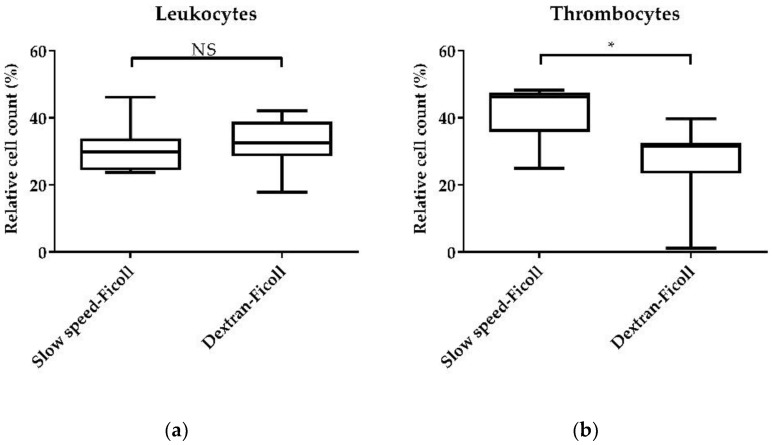
Yield of leukocytes and thrombocytes using two immune cell isolation methods, combined slow-speed-ficoll and dextran–ficoll separation. After PBMC isolation, immune cells were stained with the pan-leukocyte marker CD45 and the thrombocyte marker CD41/CD61 and subjected to flow cytometry. (**a**) Leukocytes and (**b**) thrombocytes relative to the total live cell count. A total of 20,000 vital lymphocytes were recorded on a BD Canto II flow cytometer. DAPI was used as a viability marker. Data represent seven biological replicates and two technical replicates, each in two independent experiments. A box-and-whisker plot is displayed. Significance is shown as *, *p* < 0.05. NS: not significant. Significance was analyzed using an unpaired Student’s *t*-test.

**Figure 3 animals-11-03600-f003:**
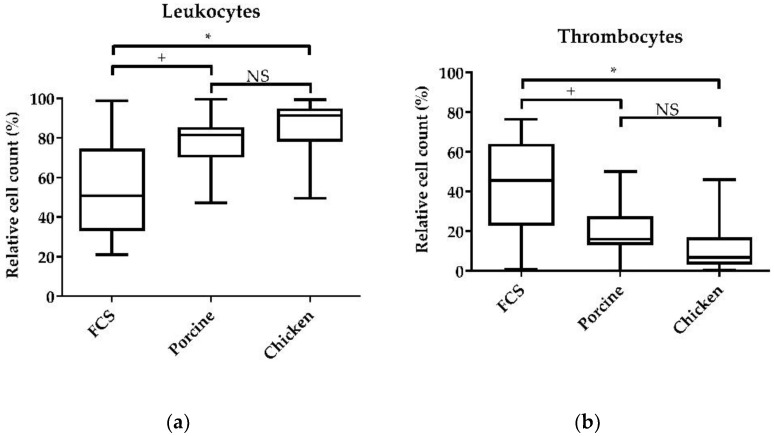
Influence of the addition of different sera to the cell culture on the survival of immune cells. Isolated PBMCs were cultured in RPMI-1640 medium with glucose, 100 U/mL penicillin, 100 µg/mL streptomycin, and either 10% chicken, 10% porcine, or 10% FCS. The relative cell counts of leukocytes and thrombocytes were assessed after 24 h of cultivation. (**a**) CD45-high leukocytes, relative to the vital lymphocyte population; (**b**) CD45-low thrombocytes relative to the vital lymphocyte population. A total of 20,000 cells were recorded on a BD Canto II flow cytometer. DAPI was used as a viability marker. Data represent seven biological replicates and two technical replicates each. A box-and-whisker plot is displayed. Significance is shown as +, *p* < 0.1; *, *p* < 0.05. NS: not significant. Significance was analyzed using an unpaired Student’s *t*-test.

**Figure 4 animals-11-03600-f004:**
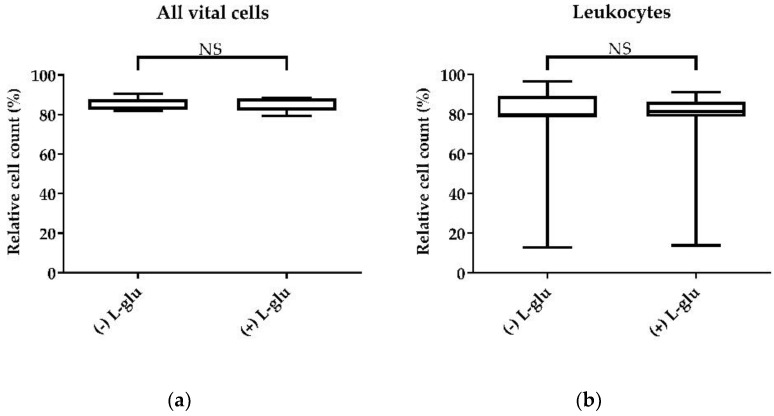
The effect of additional L-glutamine on cell viability. Isolated chicken PBMCs were cultured in RPMI-1640 medium with glucose, 100 U/mL penicillin, 100 µg/mL streptomycin, and 10% chicken serum with or without additional L-glutamine (2 mM) for 24 h. (**a**) Vital cells relative to the lymphocytes; (**b**) CD45-high leukocytes, relative to the vital lymphocyte population. A total of 20,000 cells were recorded on a BD Canto II flow cytometer. DAPI was used as a viability marker. Data represent seven biological replicates and two technical replicates each. A box-and-whisker plot is displayed. NS: not significant. Significance was analyzed using an unpaired Student’s *t*-test.

**Figure 5 animals-11-03600-f005:**
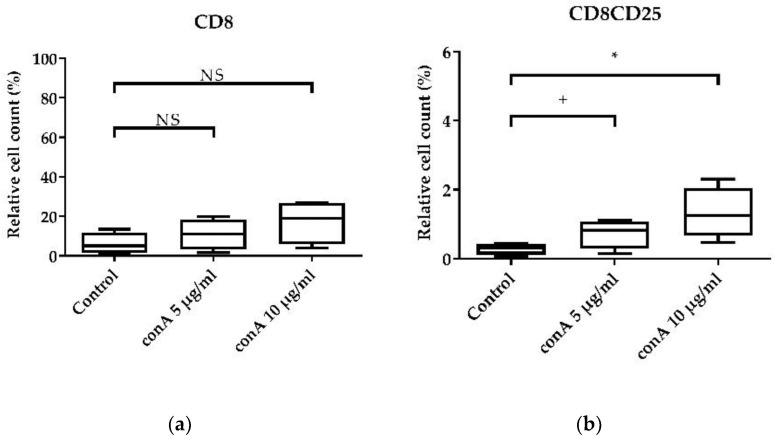
Effects of conA on CD8+ cytotoxic T-cells and CD4+ T-helper cells. The effects of two concentrations (5 µg/mL and 10 µg/mL) of conA on the activation and proliferation of cytotoxic T-cells and T-helper cells relative to the vital lymphocyte population were assessed after 24 h of cultivation. (**a**) CD8+ cytotoxic T-cells; (**b**) CD8+CD25+ activated cytotoxic T-cells; (**c**) CD4+ T-helper cells; (**d**) CD4+CD25+ activated T-helper cells. A total of 20,000 cells were recorded on a BD Canto II flow cytometer. DAPI was used as a viability marker. Data represent four biological replicates and two technical replicates each. A box-and-whisker plot is displayed. Significance is shown as +, *p* < 0.1; *, *p* < 0.05. NS: not significant. Significance was analyzed using an unpaired Student’s *t*-test.

**Figure 6 animals-11-03600-f006:**
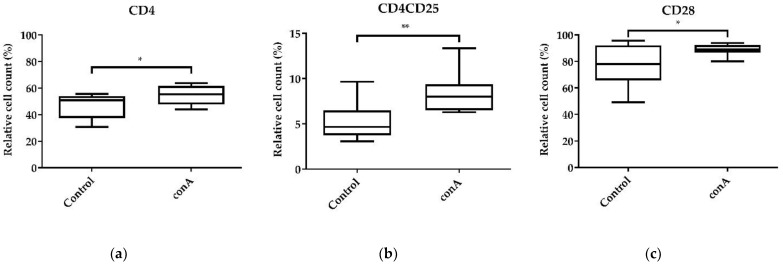
Effects of conA on activation and proliferation of CD4+ T-helper cells and CD28+ T-cells. The effect of 10 µg/mL conA on the activation and proliferation of T-helper cells relative to the vital lymphocyte population was assessed after 24 h of cultivation. (**a**) CD4+ T-helper cells; (**b**) CD4+CD25+ activated T-helper cells; (**c**) CD28+ T-cells. A total of 20,000 cells were recorded on a BD Canto II flow cytometer. DAPI was used as a viability marker. Data represent eight biological replicates and two technical replicates each. A box-and-whisker plot is displayed. Significance is shown as **, *p* < 0.01; *, *p* < 0.05. NS: not significant. Significance was analyzed using an unpaired Student’s *t*-test.

**Figure 7 animals-11-03600-f007:**
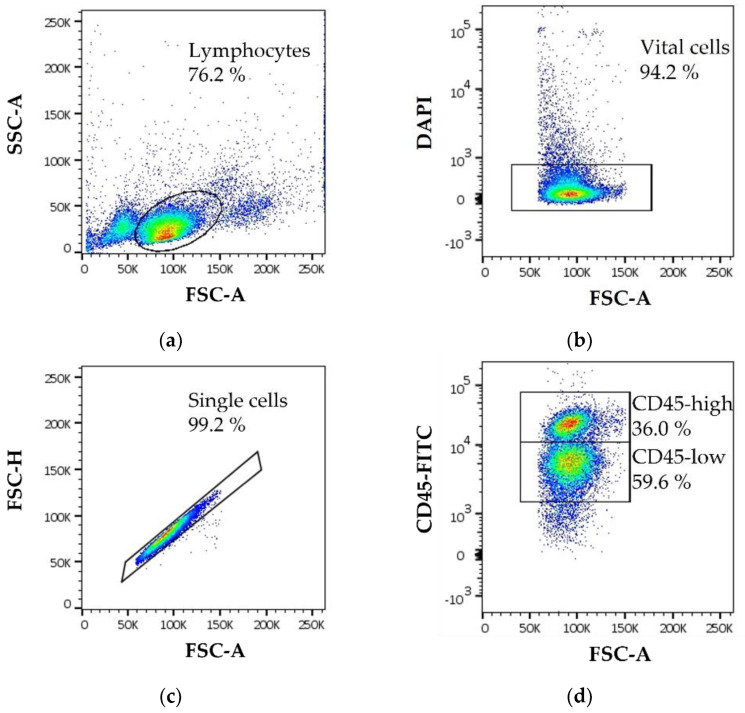
Example of the gating strategy for chicken PBMCs (**a**) *y* axes: SSC-A, *x* axes: FSC-A, the egg-shaped gate represents the lymphocyte gate; (**b**) *y* axes: DAPI, *x* axes: FSC-A, DAPI was used as a live/dead marker, the rectangle gate represents the vital lymphocyte population; (**c**) *y* axes: FSC-H, *x* axes: FSC-A, the rectangle gate represents the single cell population out of the vital lymphocyte population in B; (**d**) *y* axes: CD45-FITC, *x* axes: FSC-A, the two rectangle gates represent the CD45-high leukocyte and CD45-low thrombocyte populations out of the single cell and vital lymphocyte populations in B and C. Data represent one biological replicate. A total of 20,000 cells were recorded on a BD Canto II flow cytometer.

**Table 1 animals-11-03600-t001:** Antibodies with the conjugated fluorophore, its isotype, and the concentration used after antibody titration.

Antibody	Isotype	Final Concentration
CD45-FITC	LT40	1:50
CD41/CD61-(R)PE	11C3	1:50
CD4-SPRD	CT-4	1:50
CD8-APC	CT-8	1:50
CD25-FITC	AbD13504	1:25
CD28-PE	AV7	1:50
CD3-APC	CT-3	1:25

## Data Availability

Data supporting reported results will be provided upon request.

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
