# Peer review of "Chicken Immune Cell Assay to Model Adaptive Immune Responses In Vitro"

_animals, 2021, doi:10.3390/ani11123600_

Round 1
Reviewer 1 Report
The manuscript by Larsberg et al. described an experimental protocol to isolate chicken PBMCs for the immunological assessment. Overall, the authors did compare some variations in different steps of PBMCs isolation and culture but the authors did not include practical evidence of how their choice could affect the result of experiments. Please refer to the below comments.
- Throughout the manuscript, the authors claimed the protocol tested in this study could be used in the assessment of feed additives in the context of immunomodulatory property, but besides the feed additives, there are many things that can be used to evaluate their functions in the setting of this kind of immunological assessment. It is a lack of explanation why the authors wanted to focus on feed additives and, a clear link between immunomodulation and feed additives in the manuscript. I would suggest changing the objective and title of the manuscript.
- In chickens, the proportion of leukocytes can be largely different by age. The authors mentioned 3 to 6 week-old chickens were used but it sounds they have quite different compositions in population. Given that the maturation of the chicken immune system takes several weeks, it seems that the authors need to consider the ages of chickens as it could greatly affect the proportion of lymphocytes.
- In figure 1, the authors reported a total number of viable cells. The numbers from 40 mL (it was mentioned ‘approximately 40 mL’ in the manuscript) of blood? It needs to be from an identical amount of samples.
- Please mention in detail why the authors decide to use Na-citrate even it has shown high thrombocytes population. It seems to conflict with the authors mentioned in the manuscript (line 340-342).
-Reference no. 21 seems too old. Please change.
- No explanation of figure 6d in the manuscript.
- It is confusing how long the PBMCs were cultured. It was mentioned 1 and 3 days in the M & M section.
- In immunological assessment such as the phenotyping of lymphocytes, it seems the lymphocytes are considered more important than leukocytes. The authors choose chicken serum over FCS although the supplementation of FCS showed a higher lymphocytes population. Please explain.
- It is supposed that there is insufficient evidence to prove CD25 as an activation marker of T cells in chickens. Please provide if the authors have any other data using other surface markers (e.g. CD3).
- It looks better to provide the data on thrombocyte count in blood samples using Na-citrate (line 192-195), as well as the data in detail on titration of antibodies (line 300-303).
Author Response
Dear Reviewer,
please see the attachment.
Kind regards,
Filip Larsberg

Reviewer 2 Report
- in the Title " In vitro model for the investigation of immunomodulatory functions of feed additives in chicken ", there were "for the investigation of immunomodulatory functions of feed additives",that may attract the authors' interests. But in the contents I did not found any study for the investigation of immunomodulatory funvtions of feed additives. So the title should be changed.
-
The purpose of additional L-glutamine in Medium was not clear.
Author Response

(The authors gave the same response as above.)

Reviewer 3 Report
The manuscript In vitro model for the investigation of immunomodulatory functions of feed additives in chicken aims to establish a protocol for isolation of peripheral blood mononuclear cells to be used in culture for use in studies on the in vitro effect
immunomodulator of food additives immunomodulation. And the authors inform that the described methodology allows obtaining pure peripheral blood mononuclear cells.
However, the authors did not obtain pure PBMC, the presence of thrombocytes is approx. 27.23%.
This contamination value obtained is greater than that which can be obtained by the technique described by Gogal et al., 1997 (Gogal RM Jr, Ahmed SA, Larsen CT. Analysis of avian lymphocyte proliferation
by a new, simple, nonradioactive assay (lympho-pro). Avian Dis. 1997 Jul-Sep;41(3):714-25.)
The introduction does not cite articles relevant to the topic such as
Gogal RM Jr, Ahmed SA, Larsen CT. Analysis of avian lymphocyte proliferation by a new, simple, nonradioactive assay (lympho-pro). Avian Dis. 1997 Jul-Sep;41(3):714-25.
De Boever S, Croubels S, Demeyere K, Lambrecht B, De Backer P & Meyer E. Flow cytometric differentiation of avian leukocytes and analysis of their intracellular cytokine expression. Avian Pathology. 2010 39:1, 41-46, DOI: 10.1080/03079450903473574
Al-Khalifa H. Immunological techniques in avian studies. World's Poultry Science Journal. 2016 72:3, 573-584, DOI: 10.1017/S0043933916000532
Grasman KA. Assessing immunological function in toxicological studies of avian wildlife. Integr Comp Biol. 2002 Feb;42(1):34-42. doi:10.1093/icb/42.1.34.
Furthermore, the methodology is poorly described.
Author Response

(The authors gave the same response as above.)

Round 2
Reviewer 1 Report
Overall, the authors have addressed most of the concerns raised in the previous review. However, the manuscript still needs to be edited by deleting the things related to ‘feed additives’ as the experiments described in the manuscript are not only for the feed additives. For examples, in the abstract (line 20), keywords (line 43), introduction (line 50-55, 80, 288), and conclusions (line 477). Besides it, the manuscript is well revised.
Author Response

(The authors gave the same response as above.)

Reviewer 3 Report
The authors have made significant changes to the manuscript and I have only a few additional observations.
1) Authors should carefully review the cited references.
For example. In the sentence "Thrombocytes can be excluded in further analysis, i.e. by flow cytometry. In contrast to lymphocytes, they
express lower levels of the pan-leukocyte marker CD45 [18–19]", references 18 and 19 are not about thrombocytes.
2) Authors should change from "Blood samples were dilution 1:1" to Blood samples were dilution 1:2.
3) Authors should inform ethics committee approval for animal use.
Author Response

(The authors gave the same response as above.)
